# A Novel Algorithm for Scenario Recognition Based on MEMS Sensors of Smartphone

**DOI:** 10.3390/mi13111865

**Published:** 2022-10-30

**Authors:** Xianghong Li, Hong Yuan, Guang Yang, Yingkui Gong, Jiajia Xu

**Affiliations:** Aerospace Information Research (AIR) Institute, Chinese Academy of Sciences (CAS), Beijing 100091, China

**Keywords:** scenario recognition, feature extraction, MEMS sensors, smartphone mode, motion mode

## Abstract

The scenario is very important to smartphone-based pedestrian positioning services. The smartphone is equipped with MEMS(Micro Electro Mechanical System) sensors, which have low accuracy. Now, the methods for scenario recognition are mainly machine-learning methods. The recognition rate of a single method is not high. Multi-model fusion can improve recognition accuracy, but it needs to collect many samples, the computational cost is high, and it is heavily dependent on feature selection. Therefore, we designed the DT-BP(decision tree-Bayesian probability) scenario recognition algorithm by introducing the Bayesian state transition model based on experience design in the decision tree. The decision-tree rules and state transition probability assignment methods were respectively designed for smartphone mode and motion mode. We carried out experiments for each scenario and compared them with the methods in the references. The results showed that the method proposed in this paper has a high recognition accuracy, which is equivalent to the accuracy of multi-model machine learning, but it is simpler, easier to implement, requires less computation, and requires fewer samples.

## 1. Introduction

With the rapid development of Micro Electro Mechanical System(MEMS)sensors, the smartphone is equipped with an inertial measurement unit (IMU), barometer and magnetometer, which provide new and cheap approaches to smartphone-based pedestrian positioning services. Now, it has penetrated all aspects of people’s lives. However, because of the complex scenarios faced by the smartphone, obtaining a high-precision position based on the smartphone is still a challenge. The complex scenarios include the diversity of smartphone carrying modes, pedestrian movement modes, and the accuracy limitations of smartphone built-in sensing, which are all factors that affect the accuracy of pedestrian positioning. Contextual information is very important to the positioning system. It not only affects the types of available signals, but also provides more information for positioning, and provides an important basis for positioning methods, the selection of fusion algorithms, and failure detection. Therefore, it is necessary to identify different scenarios and choose different coping strategies for different scenarios to obtain high-precision pedestrian positioning results.

At present, pedestrian motion mode recognition is mainly divided into two research directions: one is based on image processing technology, which converts the input image or video into feature vectors, and then recognizes the motion mode [1,2]. However, it is easily infringes on personal privacy and relies heavily on light conditions [3]. The other is based on various sensors, such as accelerometers, gyroscopes, gravimeters, barometers, etc., that collect sensor data, extract various features, and classifying features. Then, various methods are used to recognize the movement pattern. Machine learning methods are mostly used for motion pattern recognition based on sensors, such as support vector machine (SVM) [4], k-nearest neighbor algorithm (KNN) [3], Gaussian naive Bayes (GNB), and artificial neural network (ANN). The average recognition success rate can reach more than 80%. For example, Sun Bingyi et al. [5] proposed a behavior recognition method based on the SC-HMM algorithm, which can classify up and down stairs and elevators with a classification accuracy of more than 80%. Jin-Shyan Lee et al. [6] proposed a threshold-based classification algorithm for carrying phone mode with an acceleration value, which is very simple and easy to achieve. Qinglin Tian et al. [7] proposed a finite state machine (FSM) to classify the smartphone mode with a classification accuracy of more than 89%.

Other scholars have tried to combine different models to improve recognition performance. Liu Bin et al. [8] combined four typical methods: k-nearest neighbor algorithm, support vector machine, naive Bayesian network, and the AdaBoost algorithm based on a naive Bayesian network to create a human activity recognition model. The optimal human activity recognition model was obtained through model decision-making, and the accuracy reached 92%. Using support vector machine (SVM) and decision tree (DT), any combination of motion state and mobile phone postures could be successfully identified [9] with an average success rate of 92.4%. Other scholars combined convolutional neural network (CNN) and long-term and short-term memory (LSTM) network to recognize walking, sitting, and lying behaviors for wearable (tied-to-the-waist) devices, with a success rate of over 96% [10,11]. In addition, some scholars used other methods to realize motion pattern recognition, such as the implicit Markov model [12,13], the sensor data interception method based on last bit matching [14], and the voting method [15]. Some scholars have also studied the influence of window length on human motion pattern recognition, to choose the optimal window length [16,17].

Ichikawa et al. [18] studied the ways people like to use mobile phones. The common locations of mobile phones are trouser pockets, clothing pockets, hand-held and so on. Scholars have explored various methods to identify the common locations, that is, to identify the location of the mobile phone. Yang et al. [19] proposed PACP (Parameters Adjustment-Corresponding-to-smartphone position), a method that is independent of the smartphone mode. It uses the SVM (support vector machine) model to identify the smartphone mode with an accuracy rate of 91%. Deng et al. [20] proposed to recognize the location of mobile phones based on accelerometer features, and tested the recognition results based on SVM, Bayesian network, and random forest. Noy et al. [21] used KNN, decision tree, and XGBoost to test and compare, and showed that XGBoost has the best recognition success rate. Wang [22] proposed a recognition method of the superimposed model, which combines the six models of AdaBoost, DT, KNN, LightGBM, SVM, and XGBoost to realize the location recognition of smartphones, and the recognition accuracy can reach 98.37%.

In general, the methods for scenarios recognition mainly focus on machine learning methods, such as SVM, CNN, KNN, etc. These methods have a low recognition accuracy rate when recognizing based on raw data, and when recognizing based on sensor features, they have a strong dependence on feature selection. The fusion of multiple models can improve the recognition accuracy, increasing the complexity of calculations and requiring a large number of samples. In addition, the calculation cost is large, and the choice of features is heavily dependent.

To solve this problem, we designed a DT-BP (decision tree-Bayesian probability) scenarios recognition algorithm by using a single model decision tree and a Bayesian state transition model, which aims at motion mode and smartphone mode. This method is more simplified, less computationally expensive, and less computationally complex, and can obtain the same recognition accuracy as the multi-model machine learning method. The contributions of this study are as follows:We designed a decoupling analysis method to analyze the relationship between e different kinds of scenario, so as determine the identification order. As the interactions of different scenario are categorised, adverse effects on scenario recognition occur. Therefore, a decoupling relationship analysis method was designed to decouple different scenario categories and determine the sequence of scenario type identification;We designed a DT-BP (decision tree-Bayesian probability) scenario recognition algorithm by using a single model decision tree and a Bayesian state transition model, which aimed at motion mode and smartphone mode. This method is more simplified, less computationally expensive, and less computationally complex, and can obtain the same recognition accuracy as the multi-model machine learning method;We designed the corresponding decision tree criteria and probability allocation method for smartphone mode and motion mode. We carried out experiments for each scenario and compared them with the methods in the references.

The rest of this paper is organized as follows: Section 2 introduces the presented algorithm, including decoupling analysis, feature extraction and scenario recognition algorithm. Section 3 shows the experimental setup, results and discussion. And finally, Section 4 concludes the paper.

## 2. Methodology

### 2.1. Decoupling Analysis of Scenario Category

It is necessary to analyze the decoupling relationship of different scenario categories to determine their independence and correlation. For example, if there are *nk* motion modes, *mk* smartphone modes, there are nk×mk kinds of situations in any combination of the two kinds of contexts. It is too complicated and redundant to identify all the combined scenarios. And as the interaction of different scenario is categorized, adverse effects on scenario recognition occur. Therefore, a decoupling relationship analysis method was designed to decouple different scenario categories and determine the sequences of scenario type recognition.

The decoupling of smartphone modes and motion modes needs to be analyzed in three parts:The correlation coefficient of the same motion mode in different smartphone modes;The correlation coefficient of different motion modes in the same smartphone mode;The correlation coefficient between different smartphone modes and different motion modes.

We used Pearson’s correlation coefficient to analyze the decoupling of the data. The calculation formula of the correlation coefficient is as follows:(1)$$r=\frac{\sum_{i=1}^{n}\left(x_{i}-\bar{x}\right)\left(y_{i}-\bar{y}\right)}{\sqrt{\sum_{i=1}^{n}{\left(x_{i}-\bar{x}\right)}^{2}}\sqrt{\sum_{i=1}^{n}{\left(y_{i}-\bar{y}\right)}^{2}}},\left\{ \begin{array}{l} \bar{x}=\sum_{i=1}^{n}x_{i}/n\\ \bar{y}=\sum_{i=1}^{n}y_{i}/n \end{array}\right.$$
where *r* is the correlation coefficient, *n* is the length of the data, $x_{i},y_{i}$ which are two different time series.

As the data sampling length in different situation is different and the data contains the periodic behavior of pedestrian movement, it was necessary to establish a time-lag series [23]. The sequence $\left(X,Y\right)=\left\{\left(x_{i},y_{i}\right),i=1,2,\ldots ,n\right\}$, after moving forward and backward by m sampling points is:(2)$$\begin{array}{l} \left(X_{t},Y_{t+m}\right)=\left\{\left(x_{i},y_{i+m}\right),i=1,2,\ldots ,n-m\right\},1\leq m<n,m\in N^{+}\\ \left(X_{t},Y_{t-m}\right)=\left\{\left(x_{i},y_{i-m}\right),i=m+1,m+2,\ldots ,n\right\},1\leq m<n,m\in N^{+} \end{array}$$

If the time-shift sequence is correlated, it must exist $m_{0}\left(1\leq \left|m_{0}\right|\leq n,\left|m_{0}\right|\in N^{+}\right)$ to maximize the correlation coefficient of $\left(X_{t},Y_{t+m_{0}}\right)$.

To decouple different scenario categories, the following analysis method was used:

(1)To avoid dependence on feature selection, raw data were selected for data analysis;

To ensure a full analysis of different scenario categories, it was necessary to exclude other factors as far as possible, such as pedestrian differences, smartphone brand differences, etc. Therefore, the window length *n* shall meet the pedestrian movement cycle, generally 0.5–1.2 s. The calculation method of window length *n* is as follows:(3)$$n>2\times \left[\frac{T_{n}}{\Delta t}\right]$$
where $T_{n}$ is the window period, generally bigger than 2 s. Δt is the sampling period, which depends on smartphone’s brand and model. [] means round numbers.

(2)To ensure the integrity of pedestrian motion cycle, the forward and backward sampling points m should be selected as:(4)$$\left[0.5\times \frac{T_{n}}{\Delta t}\right]+1\leq m<n$$(3)To ensure the analysis is not disturbed by abnormal data, the sliding window is used to calculate the correlation coefficient *r*, which is $ℜ=r_{i}\left(i=1,2,\ldots ,k\right),k=\left[N/n\right]$, and *N* is the sampling length of data. The decoupling analysis correlation coefficient is
(5)$$\bar{r}=\frac{1}{k}\sum_{i=1}^{k}r_{i}$$(4)To analyze the decoupling correlation different scenario categories, we set $r\left(x_{i,j},y_{u,v}\right)=\bar{r}$. Where *i* and *u* represent the smartphone mode, *j* and *v* represent the motion mode. $y_{u,v}$ and $y_{u,v}$ are two kinds of scenario. To obtain the analysis result we needed to analyze three situations as follows:(6)$$\left\{ \begin{array}{l} i=u,\;\;\;j\neq v\\ i\neq u,\;\;\;j=v\\ i\neq u,\;\;\;j\neq v \end{array}\right.$$

The test results are shown in Table 1, which gives the correlation calculation results of a total of nine scenarios composed of three motion modes and three smartphone modes. The raw data include GNSS sensor, accelerometer, gyroscope, magnetometer, barometer, and Bluetooth. 

According to the test results in Table 1, the decoupling correlation can be summarized as follows:(7)$$\left\{ \begin{array}{l} 0.3<r\left(x_{i,j},y_{u,v}\right)<0.4,when\;i=u,\;\;\;j\neq v\\ r\left(x_{i,j},y_{u,v}\right)>0.5,when\;i\neq u,\;\;\;j=v\\ r\left(x_{i,j},y_{u,v}\right)<0.3,when\;i\neq u,\;\;\;j\neq v \end{array}\right.$$

From Formula (7) we know that the correlation between different motion modes has a certain correlation under the same smartphone mode. In the case of different smartphone modes, the correlation of the same motion mode is greater than 0.5. The correlation between different smartphone modes and different motion modes was very low, and less than 0.3. So, smartphone modes have little influence on motion mode recognition. On the contrary, motion modes have a great influence on smartphone mode recognition.

According to the above analysis, during scenario recognition, we can recognize the motion mode first. And when this is determined, the smartphone mode is recognized secondly.

### 2.2. Feature Extraction

In this paper, we extracted features from different sensor data in both the time domain and frequency domain, respectively. The time-domain refers to the extraction of the mathematical-statistical characteristics of the sensor measurement data in a certain window length, such as variance, mean, amplitude, etc. The frequency domain refers to the calculation of the Fourier transform and frequency domain entropy of the sensor measurement data in a certain window length. Then, the features in the frequency domain were extracted, such as dominant frequency, energy, frequency difference, etc.

As shown in Table 2, the time-domain features extracted from active sensors were used in this paper. Where $n_{1},n_{2},n_{3},n_{4},n_{5},n_{6}$ is the length of the data window, $x_{i},y_{i},z_{i}$ is the sampling data, $\bar{x}$ is the mean value, *i,j,k* are the times at different sampling points, $s_{k}$ is the value that the data in the window length $n_{4}$ is greater and smaller than the threshold value $thre_{zero},thre_{-zero}$.

The height gradient value [25,26] is calculated by the raw data of barometer as:(8)$$dh=h-h_{0}=18400\cdot \left(1+\frac{t}{273.15}\right)\cdot \mathrm{lg}\frac{p_{0}}{p}$$
where *t* is the temperature, and the unit is °C. $p_{0}$ is the reference air pressure and *p* is the output of the barometer.

### 2.3. Scenario Recognition Algorithm

#### 2.3.1. Design of Scenario Recognition Algorithm Based on DT-BP

The decision tree (DT) establishes the nodes by exploring the high-value data features in the overall data and constructs the branches of the tree according to the required research contents. With repeatedly establishing the branch nodes, the classification results and decision set contents are displayed with the tree structure [27,28]. The decision tree has the advantages of low computational complexity and is insensitive to missing content in the middle. It can handle irrelevant feature data [29]. Decision trees also have shortcomings, such as low detection accuracy and the work needed for the preprocessing of time-sequential data. In the actual environment, due to the complexity of the environment, the different interference, the performance of different devices, the error accumulation of the sensor itself, etc., the error in the identification process is so relatively large that the usability is not high. To deal with this problem, we designed a context recognition method based on the decision tree and Bayesian state transition probability (decision tree-Bayesian probability, DT-BP).

Bayes theory is a common method in model decision-making. The basic idea is to know the conditional probability density parameter expression and a priori probability, convert the formula into a posteriori probability, and finally use a posteriori probability for decision classification.

If P(A) is a priori probability or edge probability of *A*. the conditional probability of *A* after the occurrence of *B* is P(A|B), which is called the posterior probability of *A*. P(B|A) is the conditional probability of *B* after the occurrence of *A*, which is called the posterior probability of *B*. P(B) is the prior probability of *B* [30]. Then
(9)$$P\left(B|A\right)=\frac{P\left(A|B\right)P\left(B\right)}{P\left(A\right)}$$

For the situation recognition in this paper, we suppose $T_{i}$ as the situation category, and $T_{i}\in \Gamma \left(b_{1},b_{2},\ldots ,b_{u}\right)$. Where *U* is the number of situations. We assume $S_{f}$ as the feature quantity set, and $S_{f}=\left\{a_{1},a_{2},\ldots ,a_{V}\right\}$. Where *V* is the number of features. If the features of $S_{f}$ all belong to $T_{i}$, the probability is $P\left(T_{i}|S_{f}\right)$. When it is satisfied $P\left(T_{k}|S_{f}\right)=\underset{i=1,\cdots ,K}{\max}P\left(T_{i}|S_{f}\right)$, it is considered $S_{f}\in T_{k}$, which means the recognition is successful. Therefore, we only need to calculate $P\left(T_{i}|S_{f}\right)$ to recognize context. The Formula (9) is changed to:(10)$$P\left(T_{i}|S_{f}\right)=\frac{P\left(T_{i}\right)P\left(S_{f}|T_{i}\right)}{P\left(S_{f}\right)}$$

To obtain the conditional probability of the scenario $T_{i}$, we need to calculate $P\left(S_{f}\right)$, $P\left(T_{i}\right)$ and $P\left(S_{f}|T_{i}\right)$.

The principle of probability allocation in this article: the number of features quantities is *V*, and the probability of each feature quantity is the same, which means $P\left(S_{f}\right)$ is a constant. So $P\left(T_{i}|S_{f}\right)$ is the largest when $P\left(T_{i}\right)P\left(S_{f}|T_{i}\right)$ is the largest. That is
(11)$$P\left(T_{i}|S_{f}\right)\propto P\left(T_{i}\right)P\left(S_{f}|T_{i}\right)$$
where $P\left(T_{i}\right)$ is the state probability. Its value at the current moment is related to the number of scenarios to be detected and the probability of the previous moment. It is independently designed according to different scenario categories and the number of scenarios *U*. $P\left(S_{f}|T_{i}\right)$ is the conditional probability of the feature vector $S_{f}$, which is obtained from DT rules. The obtaining algorithm designed in this paper is as follows:(12)$$P\left(S_{f}|T_{i}\right)=\frac{\sum_{j=1}^{K_{i}}c_{j}}{K_{i}}$$
where $K_{i}$ represents the number of features related to the category $T_{i}$. $c_{j}$ is the judgment value of each feature. If the judgment condition is met, it is $c_{j}=1$, otherwise is $c_{j}=0$.

#### 2.3.2. Recognition of Smartphone Mode Based on DT-BP

(1)Algorithm design based on decision tree

To realize the recognition of different smartphone modes, it is necessary to detect the transformation process between different smartphone modes, which determines whether the smartphone mode is transforming or fixed. When it is in the fixed position, the different smartphone mode is determined. There are two parts of mobile phone location recognition: transformation recognition and current location recognition. In this paper, we took six common smartphone modes [18] as examples to design the specific decision tree, including texting, calling, pants front pocket, clothes pocket, pants back pocket and hand swing, as shown in Figure 1.

According to the data characteristic analysis and feature analysis, the variance of the first-order norm of acceleration was used as the criterion for judging the position change of the mobile phone. The first-level decision criterion for the decision tree is:(13)$$\left\{ \begin{array}{l} if\mathrm{var}\left(ax\right)>Thre\&\mathrm{var}\left(ay\right)>Thre\&\mathrm{var}\left(az\right)>Thre,changing=1\\ others,changing=0 \end{array}\right.$$
where *Thre* is the threshold. When changing=1, the location is changing, otherwise it is not.

When the position is fixed, it is necessary to determine whether there is periodic oscillation, which means there are other periodic motions besides walking and swinging with the pedestrian. We use the second main frequency amplitude of the acceleration amplitude to determine it. The second-level decision criterion of the decision tree is:(14)$$\left\{ \begin{array}{l} if(A_{mf2}>Thre_{mf2}),per=1;\\ others,per=0 \end{array}\right.$$
where $Thre_{mf2}$ is the threshold. When per=1, the position of the mobile phone has periodic movement, otherwise it is not. If the smartphone location with periodic movement, the judgment criterion is as follows:(15)$$\left\{ \begin{array}{l} if(F_{mf}>Thre_{mf}),swpp=1;\\ others,swpp=0 \end{array}\right.$$
where $Thre_{mf}$ is the threshold. When swpp=1, the position is pants pocket(pp), otherwise it is swinging. As the features of the front pants pocket(fpp) and back pants pocket(bpp) is similar, there is a new branch for them, and the decision criterion is:(16)$$\left\{ \begin{array}{l} if\;\;\;\frac{1}{M}\sum_{i=1}^{M}\left(ax-az\right)>Thre_{qh},fpp=1\\ others,bpp=1 \end{array}\right.$$
where *M* is the window length, $Thre_{qh}$ is the threshold.

To recognize the fixed smartphone mode without periodic motion, we used the first-order norm of acceleration, peaks, and wave as the features. The determination rule for designing a fixed smartphone mode decision tree is:(17)$$\left\{ \begin{array}{l} text:\begin{array}{ccc} \left|ax\right|<Thre1,\;\;\; & \left|ay\right|<Thre1,\; & az_{\min}>Thre2 \end{array}\;\;\;\;\;\;\\ call:\begin{array}{ccc} ax_{\max}<-Thre1, & ay_{\min}>Thre2,\;\;\; & \left|az\right|<Thre1 \end{array}\\ pocket:\begin{array}{ccc} ax_{\min}>Thre2, & ay_{\max}<-Thre1, & \left|az\right|<Thre1 \end{array} \end{array}\right.$$
where *Thre*1, *Thre*2, *Thre*3 are thresholds.

(2)Probabilistic design based on DT-BP method

According to the design of the DT-BP method, we needed to design $P\left(T_{i}\right)$ and $P\left(S_{f}|T_{i}\right)$ to calculate the probability distribution principle. The number of scenarios *U* is 7, including 6 smartphone positions and the change process. $P\left(T_{i}\right)$ is designed as shown in Table 3. According to the design of the decision tree, the number of features *V* is 10. The design of $K_{i}$ in Formula (12) is as follows in Table 4.

#### 2.3.3. Recognition of Motion Mode Based on DT-BP

(1)Design of motion mode recognition algorithm based on decision tree

In this paper, the recognition algorithm is designed by taking the motion modes of static, walking, turning, going upstairs and downstairs, escalator, and elevator as examples. According to the analysis of the extracted features, the dynamic and static are distinguished by the acceleration variance. We used positive and negative zero-crossing rates of acceleration air pressure gradients as the features of static, elevator and escalator recognition. Dynamic motion includes walking on the ground, going upstairs, going downstairs and turning. Walking, turning, going upstairs and downstairs are coupled movements, that is, turning and going up and down stairs also have walking movements. Therefore, the main work was to distinguish turning and going up and down stairs from walking. The amplitude of angular velocity is used to recognize turning. We used the auto-correlation coefficient, Fourier dominant frequency, and elevation gradient to recognize going up and down stairs and walking, as shown in Figure 2.

The process of motion mode recognition based on the decision tree method is as follows:

(a)If var(a) is greater than the threshold $Thre_{a}$, it is considered dynamic, otherwise, it is static.(b)When it is static, it is necessary to distinguish between static and elevator and escalator. We use the zero-crossing rate of acceleration amplitude as judgment, and the judgment criterion is:(18)$$\left\{ \begin{array}{l} flag_{el}=1,if\;\left(p_{+}\left(a_{k}\right)>Thre_{+el}\&p_{-}\left(a_{k}\right)<Thre_{-el}\right)\\ flag_{el}=-1,if\;\left(p_{-}\left(a_{k}\right)>Thre_{+el}\&p_{+}\left(a_{k}\right)<Thre_{-el}\right)\\ flag_{el}=0,otherwise \end{array}\right.$$
where $a_{k}=svm_{a}-g$, $svm_{a}$ is the amplitude of acceleration and *g* is the acceleration of gravity. $p_{+}\left(a_{k}\right)$ and $p_{-}\left(a_{k}\right)$ is the positive and negative zero-crossing rate. $Thre_{+el}$ and $Thre_{-el}$ are the zero-crossing judgment threshold. If $flag_{el}\neq 0$, it is going up or down elevator.(c)When the motion is the elevator, we judge whether going up or down. If $flag_{el}=1$ and the previous state is not the elevator, the state is going up elevator. If $flag_{el}=1$ and the previous state is the elevator, the state is going down elevator. If $flag_{el}=-1$ and the previous state is not the elevator, the state is going down elevator. If $flag_{el}=-1$ and the previous state is the elevator, the state is going up elevator.(d)If $flag_{el}=0$, it is further recognized whether it is an escalator, and the recognition criteria are as follows:(19)$$\left\{ \begin{array}{l} flag_{es}=-1,if\;\left(p_{+}\left(dbro\right)>Thre_{+es}\&p_{-}\left(dbro\right)<Thre_{-es}\right)\\ flag_{es}=1,if\;\left(p_{-}\left(dbro\right)>Thre_{+es}\&p_{+}\left(dbro\right)<Thre_{-es}\right)\\ flag_{es}=0,otherwise \end{array}\right.$$
where dbro is the pressure gradient, $p_{+}\left(dbro\right)$ and $p_{-}\left(dbro\right)$ are the positive and negative zero-crossing rates of the pressure gradient. $Thre_{+es}$ and $Thre_{-es}$ are the thresholds. When $flag_{es}=0$, it is the static state. If $flag_{es}=1$, the escalator is going up. If $flag_{es}=-1$, it is the elevator is going down.(e)When pedestrians are in a dynamic state, we mainly distinguish turning, stairs, and walking. The angular velocity amplitude is used to recognize turning. When ${\mathrm{svm}}_{w}>Thre_{w}$, it is turning.(f)We use the auto-correlation coefficient and Fourier transform to distinguish stairs and walking. The auto-correlation coefficient at the offset *k* = 2 and *k* = 4 as the judgment value, and the main frequency of the Fourier transform as the judgment condition. The criterion is as follows:(20)$$\left\{ \begin{array}{l} flag_{st}=1,if\;\left(\left|R_{a}\left(2\right)\right|>\left|R_{a}\left(4\right)\right|\&F_{a}>Thre_{fa}\right)\\ flag_{st}=0,otherwise \end{array}\right.$$
where $R_{a}\left(2\right)$ is the correlation coefficient of the acceleration amplitude offset *k* = 2. $F_{a}$ represents the main frequency of the Fourier transform. $Thre_{fa}$ is the main frequency judgment threshold. If $flag_{st}=1$, it is up and downstairs, otherwise it is walking on level ground.(g)When pedestrians are going up and down the stairs, we use the height gradient value to make judgments. The criterion is as follows:(21)$$\left\{ \begin{array}{l} flag_{h}=1,if\;\left(dh>Thre_{h}\right)\\ flag_{h}=-1,if\;\left(dh<Thre_{-h}\right) \end{array}\right.$$
where $Thre_{h}$ and $Thre_{-h}$ are the judgment thresholds of the height gradient. If $flag_{h}=1$, it is going up the stairs. If $flag_{h}=-1$, it is going down the stairs.

(2)Probabilistic design based on DT-BP method

For the nine motion modes in the above example, the probability $P\left(T_{i}\right)$ of each state in the DT-BP method is determined according to the previous state. The transition probability between each motion mode is shown in Table 5. According to Figure 2 and the decision tree rules, the number of features *V* is 12, as shown in Table 6 for the design $K_{i}$ in Formula (12).

## 3. Experimental

### 3.1. Experimental Setup

To understand the effectiveness and limitations of our proposed Scenario recognition algorithm, we conducted an implementation on Android to collect data. During the experiment, we collected data using an Android smartphone (Huawei mate 8, whose parameters are shown in Table 7), which was equipped with a three-axis accelerometer and a three-axis gyroscope. We evaluated the proposed method in six common smartphone modes (texting, calling, pants front pocket, clothes pocket, pants back pocket and hand swing) and nine natural motion modes (static, walking, turning, upstairs and downstairs, up escalator, down escalator, up elevator and down elevator). The threshold of smartphone mode and motion mode in the experimental are shown in Table 8. We compared the proposed algorithm with state-of-the-art algorithms.

### 3.2. Experimental Results of Smartphone Mode Recognition

To test the algorithm of smartphone mode recognition, we collect text, calling, front pants pocket, back pants pocket, clothes pocket, and hand-held positions, respectively. There was a total of 9901 epoch data, including 568 epoch data in the changing, 2691 epoch data in texting, 1236 epoch data in calling, 2095 epoch data in the front pants pocket 1512 epochs in the back pants pockets, 1561 epochs in clothes pocket and 238 epochs in swing.

The accuracy of smartphone mode recognition is shown in Table 9. The recognition accuracy was calculated with the following formula:(22)$$err=\frac{Num_{reco}}{Num_{true}}\times 100\%$$
where $Num_{reco}$ is the number of epochs where the state recognition result is the same as the state to be recognized. $Num_{true}$ is the actual number of epochs in the state to be recognized. The results were similar to Table 9 in many tests. The average recognition accuracy was 99.06%, the lowest accuracy was 96.48%, and the recognition accuracy of all positions was greater than 96%. The main reason is that the smartphone mode changing was followed by various fixed smartphone modes. The error of various smartphone modes will be reflected in the smartphone mode changing.

The comparison of the recognition accuracy of the algorithm in this paper with other different algorithms [20,31] is shown in Table 10. From the table, the random forest has the highest accuracy when texting and calling. DT-BP has the highest success rate in hand-held. The average accuracy in this paper is slightly higher by 0.5%. Compared with other methods based on machine learning, the DT-BP proposed in this paper has a slightly higher recognition accuracy, and takes 0.51 s in total, while the random forest takes 8.34 s. The algorithm in this paper greatly reduces the operation time and ensures the recognition success rate.

### 3.3. Experimental Results of Motion Mode Recognition

We collected static, walking, up and down stairs, elevator up and down, and escalator up and down data, respectively, to test DT-BP. There was a total of 6789 epoch data, including 3239 epochs in static, 2005 epochs in walking, 173 epochs in up the stairs, 165 epochs in down the stairs, 317 epochs in up the elevator, 319 epochs in down the elevator, 145 epochs in up the escalator, and 426 epochs in down the escalator. The recognition accuracy was calculated by the Equation (22).

The accuracy of motion recognition is shown in Table 11, which was similar to many tests. The average recognition accuracy of DT-BP was 97.3%, and the lowest was 88.73%, which is down escalator. The main reason is that there is a parallel stage of the escalator when getting on the escalator and preparing to get off the escalator. During this period, the speed is uniform. It is difficult to recognize as the speed is submerged in the noise of acceleration. The accuracy of the elevator going up and down and turning is the highest, mainly because the acceleration change characteristics are obvious in these states. Since the intermediate transition state of each motion process is static, it will be considered static when detecting errors in the other eight states. Therefore, there will be more states of detecting errors at rest.

To further analyze DT-BP proposed in this paper, it is compared with other algorithms [9,10,32,33,34], as shown in Figure 3. The recognition accuracy using a single machine learning model was relatively lower. For example, SVM and KNN are both more than 80%. The recognition success rate using multiple models will significantly improve with increases in algorithm complexity and calculation cost. The accuracy of DT-BP is the same as the method of machine learning using multiple models, and the computational complexity and computational cost are significantly reduced.

## 4. Conclusions

At present, the methods for scenario recognition are mainly machine-learning methods. The recognition accuracy of a single model is not high. Multi-model fusion can improve recognition accuracy. However, the computational cost is high, and it is heavily dependent on feature selection. We mainly focused on two types of contexts: sports mode, and mobile phone location to design a DT-BP recognition algorithm by introducing the Bayesian state transition model based on experience design into the decision tree. It is more simplified and easier to implement and has less computation and lower computational complexity. In addition, it can obtain the same recognition accuracy as the multi-model machine learning method.

## Figures and Tables

**Figure 1 micromachines-13-01865-f001:**
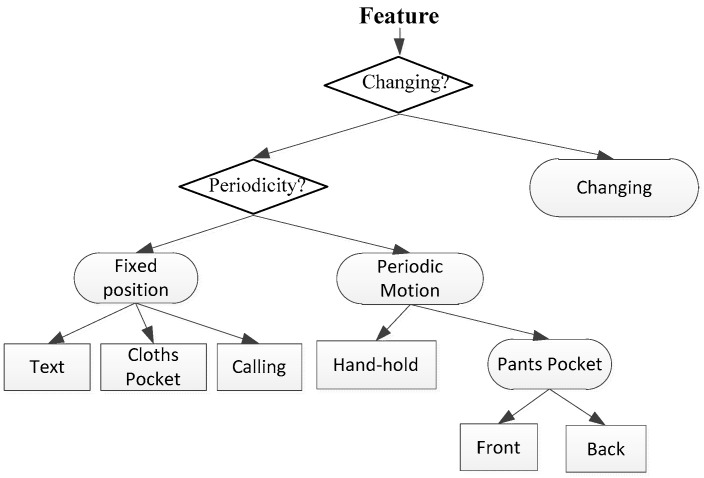
The example of decision tree for smartphone modes recognition.

**Figure 2 micromachines-13-01865-f002:**
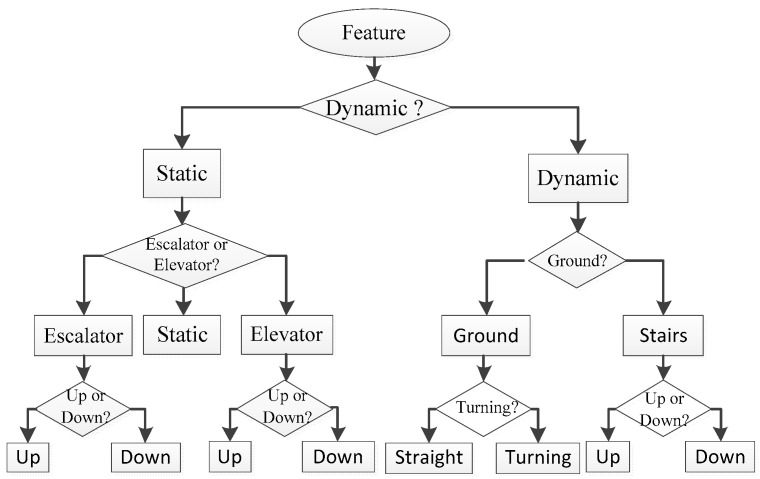
The example of decision tree for motion modes recognition.

**Figure 3 micromachines-13-01865-f003:**
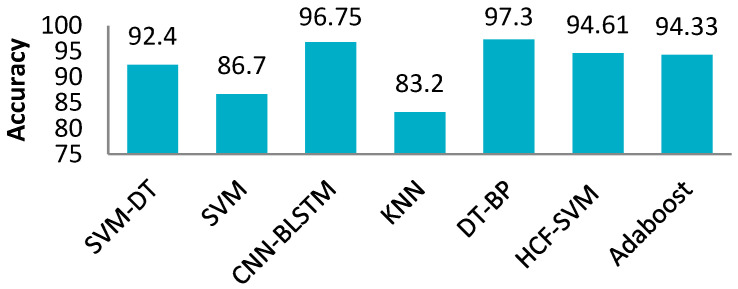
Comparison of motion mode recognition accuracy for different methods.

**Table 1 micromachines-13-01865-t001:** Pearson correlation coefficients of different motion modes and different smartphone modes.

		Wa			US			UE		
		Calling	Texting	Cloths Pocket	Calling	Texting	Cloths Pocket	Calling	Texting	Cloths Pocket
Wa	Calling	0.77	-	-	-	-	-	-	-	-
	Texting	0.56	0.64	-	-	-	-	-	-	-
	Cloths pocket	0.55	0.53	0.77	-	-	-	-	-	-
US	Calling	0.35	0.18	0.21	0.68	-	-	-	-	-
	Texting	0.27	0.34	0.25	0.55	0.80	-	-	-	-
	Cloths pocket	0.27	0.24	0.33	0.51	0.52	0.67	-	-	
UE	Texting	0.32	0.21	0.23	0.26	0.26	0.21	0.71	-	-
	Cloths pocket	0.21	0.33	0.28	0.18	0.33	0.25	0.52	0.86	-
	Texting	0.21	0.25	0.35	0.22	0.15	0.31	0.65	0.55	0.72

Wa is Walking, US is upstairs, UE is up elevator.

**Table 2 micromachines-13-01865-t002:** Feature extraction in time domain and frequency domain [24].

Feature	Definition
First-order norm	${\left\|x\right\|}_{1}=\left|x_{i}\right|$
Peaks	$a_{\max}=\max(x_{i}),i=1,\ldots ,n_{1}$
Wave	$a_{\min}=\min(x_{i}),i=1,\ldots ,n_{1}$
Difference of Peaks and Wave	$a_{mm}=a_{\max}-a_{\min}$
Mean	$\bar{x}={\frac{1}{n}}_{2}\sum_{i=1}^{n}x_{i}$
Variance	$\sigma^{2}=\frac{1}{n_{3}}\sum_{i=1}^{n_{3}}{\left(x_{i}-\bar{x}\right)}^{2}$
Amplitude	$\mathrm{svm}=\sqrt{x_{i}^{2}+y_{i}^{2}+z_{i}^{2}}$
Zero-Crossing Rate	$p_{0}=\frac{s_{k}}{n_{4}}$, $\left\{ \begin{array}{l} s_{k}=s_{k}+1,if\left(x_{i}>thre_{zero}\right)\\ s_{-k}=s_{-k}+1,if\left(x_{i}<thre_{-zero}\right) \end{array}\right.$
Gradient	$dx=x_{k+i}-x_{k}$
correlation coefficient	$r_{k}=\frac{\sum_{i=1}^{n_{5}-k}\left(x_{i}-\bar{x}\right)\left(x_{i+k}-\bar{x}\right)}{\sum_{i=1}^{n_{5}}{\left(x_{i}-\bar{x}\right)}^{2}}$
Fourier Transform	$X\left(k\right)=\sum_{i=0}^{n-1}x\left(i\right)W_{n_{6}}^{ki},k=0,1,\ldots ,n-1,W_{n_{6}}=e^{-j\frac{2\pi }{n_{6}}}$

**Table 3 micromachines-13-01865-t003:** Smartphone mode transition probability allocation.

	Changing	Text	Call	Fpp	Pocket	Bpp	Swing
Changing	1/7	1/7	1/7	1/7	1/7	1/7	1/7
Text	1/2	1/2	-	-	-	-	-
Call	1/2	-	1/2	-	-	-	-
Fpp	1/2	-	-	1/2	-	-	-
Pocket	1/2	-	-	-	1/2	-	-
Bpp	1/2	-	-	-	-	1/2	-
Swing	1/2	-	-	-	-	-	1/2

**Table 4 micromachines-13-01865-t004:** Smartphone mode related feature quantity allocation.

	Changing	Text	Call	Fpp	Pocket	Bpp	Swing
$K_{i}$	3	7	7	6	7	6	5

**Table 5 micromachines-13-01865-t005:** Motion mode transition probability allocation.

	Static	Wa	US	DS	UE	DE	UC	DC	Turning
Static	1/8	1/8	1/8	1/8	1/8	1/8	1/8	1/8	-
Wa	1/9	1/9	1/9	1/9	1/9	1/9	1/9	1/9	1/9
US	1/4	1/4	1/4	-	-	-	-	-	1/4
DS	1/4	1/4	-	1/4	-	-	-	-	1/4
UE	1/3	1/3	-	-	1/3	-	-	-	-
DE	1/3	1/3	-	-	-	1/3	-	-	-
UC	1/3	1/3	-	-	-	-	1/3	-	-
DC	1/3	1/3	-	-	-	-	-	1/3	-
Turning	1/5	1/5	1/5	1/5	-	-	-	-	1/5

Wa is Walking, US is upstairs, DS is down stairs, UE is up elevator, DE is down elevator, UC is up escalator, DC is down escalator.

**Table 6 micromachines-13-01865-t006:** Motion mode related feature quantity allocation.

	Static	W	US	DS	UE	DE	UC	DC	Turning
$K_{i}$	5	5	6	6	4	4	6	6	2

**Table 7 micromachines-13-01865-t007:** Huawei mate 8 smartphone sensor and related parameters.

Sensors	Type	Parameters
GNSS sensor	--	Support GPS, A-GPS, GLONASS and BDS
accelerometer	LSM330	Sensitivity: 0.0095768068 m/S^2^; Measurement Range: 78.4532012939 m/S^2^
gyroscope	LSM330	Sensitivity: 0.0012217305 rad/s; Measurement Range: 34.9065856934 rad/s
magnetometer	AK09911	Sensitivity: 0.0625 μT; Measurement Range: 2000 μT
barometer	AirPress sensor Rohm, BM1383	Sensitivity: 0.0099999998 hPa; Measurement Range: 1100 hPa
Bluetooth	--	4.2 + BLE

**Table 8 micromachines-13-01865-t008:** The threshold of smartphone mode and motion mode.

Smartphone Mode		Motion Mode	
Threshold	Number	Threshold	Number
Thre	2	$Thre_{a}$	1.5
$Thre_{mf2}$	1.5	$Thre_{+el}$	0.9
$Thre_{mf}$	10	$Thre_{-el}$	0.05
$Thre_{qh}$	1	$Thre_{+es}$	0.9
*Thre*1	6	$Thre_{-es}$	0.05
*Thre*2	6	$Thre_{fa}$	14
*Thre*3	8	$Thre_{h}$	0.1
		$Thre_{-h}$	−0.1

**Table 9 micromachines-13-01865-t009:** Smartphone mode recognition results.

	Changing	Text	Call	Fpp	Pocket	Bpp	Swing
Changing	96.48%	0.18%	0	0	2.64%	0	0.70%
Text	2.04%	97.96%	0	0	0	0	0
Call	1.13%	0	98.87%	0	0	0	0
Fpp	0.29%	0	0	99.71%	0	0	0
Pocket	0	0	0	0	100%	0	0
Bpp	0.40%	0	0	0	0	99.6%	0
Swing	1.68%	0	0	0	0	0	98.32%

**Table 10 micromachines-13-01865-t010:** Comparison of smartphone mode recognition accuracy for different methods.

	Text	Call	Pocket	Swing	Average
DT-BP	97.2%	98.0%	99.77%	98.32%	98.32%
SVM	97.96%	95.8%	95.2%	98.1%	96.765%
Bayesian Network	94.5%	97.2%	92.2%	93.6%	94.375%
Random Forest	99.0%	98.2%	97.1%	96.9%	97.8%

**Table 11 micromachines-13-01865-t011:** Motion mode recognition results.

	Static	W	US	DS	UE	DE	UC	DC	Turning
Static	96.10%	2.07%	0.27%	0.24%	0	0.03%	0.03%	1.25%	0
W	0.17%	99.83%	0	0	0	0	0	0	0
US	0	0	100%	0	0	0	0	0	0
DS	0	0	0	100%	0	0	0	0	0
UE	0.95%	0	0	0	99.05%	0	0	0	0
DE	1.88	0	0	0	0	98.12%	0	0	0
UC	6.46%	0	0	0	0	0	93.54%	0	0
DC	11.27%	0	0	0	0	0	0	88.73%	0
Turning	0	0	0	0	0	0	0	0	100%

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
