# Peer review of "A Novel Algorithm for Scenario Recognition Based on MEMS Sensors of Smartphone"

_micromachines, 2022, doi:10.3390/mi13111865_

Round 1

Reviewer 1 Report

This paper proposed a Decision Tree-Bayesian Probability (DT-BP) context recognition method by using a single model decision tree and a Bayesian state transition model, which aims at two types of situational information of sports mode and different smartphone mode. The subject is very interesting. However, several comments are provided as follows.

1) In the title, what does the “Scenario Recognition” mean? 

Is the scenario the human activity? sports mode? motion mode? smartphone mode? or mobile phone location? The authors should define them more clearly.

2) The reviewer is unsure about the novelty of this work. Using the smartphone posture/position to improve the human activity/motion recognition is not a new idea. The authors should explain the contributions more clearly.

2) In Sec. 1, for the combination of smartphone location/mode/posture to improve the motion recognition, the following references could be studied and cited.

[a] Safyan, M., Ul Qayyum, Z., Sarwar, S., Iqbal, M., Garcia Castro, R., & Al‐Dulaimi, A. (2019). Ontology evolution for personalised and adaptive activity recognition. IET Wireless Sensor Systems, 9(4), 193-200.

[b] Lee, J. S., & Huang, S. M. (2019). An experimental heuristic approach to multi-pose pedestrian dead reckoning without using magnetometers for indoor localization. IEEE Sensors Journal, 19(20), 9532-9542.

[c] Tian, Q., Salcic, Z., Kevin, I., Wang, K., & Pan, Y. (2016). A multi-mode dead reckoning system for pedestrian tracking using smartphones. IEEE Sensors Journal, 16(7), 2079-2093.

[d] Strozzi, N., Parisi, F., & Ferrari, G. (2018). Impact of on‐body IMU placement on inertial navigation. IET Wireless Sensor Systems, 8(1), 3-9.

3) In Sec. 2.3. Scenario Recognition Algorithm, does the recognition results (2.3.2) of the smartphone mode affect the recognition of motion mode (2.3.3)? If so, how?

4) In Sec. 3.3, Figure 3, how do you obtain the results from other algorithms [8, 9, 30-32]? By reimplementing them? Or directly from the papers?

Author Response

Point 1:  In the title, what does the “Scenario Recognition” mean? 

Is the scenario the human activity? sports mode? motion mode? smartphone mode? or mobile phone location? The authors should define them more clearly.

Response 1: With the rapid development and wide popularization of MEMS sensors, which is equipped on smartphone, pedestrian location service has penetrated all aspects of people's lives. However, the diversity of smartphone modes(such as call, text, et al.), the diversity of pedestrian motion modes(such as walking, elevator, et al.), and the diversity of complex environments( such as indoor, outdoor, et al.) are the scenario affecting pedestrian positioning, which is all called the scenario. For example, different pedestrian motion modes require different motion constraints and different dynamic models in a filter. By recognizing different scenarios, we could design different positioning algorithms, different combination models, different error estimation methods, and different fusion data sources to obtain high-precision pedestrian positioning results. In this paper, we focus on motion mode recognition and smartphone mode recognition.

Point 2: The reviewer is unsure about the novelty of this work. Using the smartphone posture/position to improve the human activity/motion recognition is not a new idea. The authors should explain the contributions more clearly.

Response 2 : Firstly, we design a decoupling analysis method to analyze the relationship between smartphone mode and motion mode to determine the recognition order of different scenario categories, instead of Using the smartphone posture/position to improve the human activity/motion recognition. Secondly, our new idea is to design a decoupling analysis method to achieve the recognition sequence of different situation categories and simplify the recognition categories in the recognition process.

Point 3: In Sec. 1, for the combination of smartphone location/mode/posture to improve the motion recognition, the following references could be studied and cited.

[a] Safyan, M., Ul Qayyum, Z., Sarwar, S., Iqbal, M., Garcia Castro, R., & Al‐Dulaimi, A. (2019). Ontology evolution for personalised and adaptive activity recognition. IET Wireless Sensor Systems, 9(4), 193-200.

[b] Lee, J. S., & Huang, S. M. (2019). An experimental heuristic approach to multi-pose pedestrian dead reckoning without using magnetometers for indoor localization. IEEE Sensors Journal, 19(20), 9532-9542.

[c] Tian, Q., Salcic, Z., Kevin, I., Wang, K., & Pan, Y. (2016). A multi-mode dead reckoning system for pedestrian tracking using smartphones. IEEE Sensors Journal, 16(7), 2079-2093.

[d] Strozzi, N., Parisi, F., & Ferrari, G. (2018). Impact of on‐body IMU placement on inertial navigation. IET Wireless Sensor Systems, 8(1), 3-9.

Response 3: Reference [a] is mainly about activity recognition (such as making tea, making coffee, et al.).  While we focus on motion mode(such as walking, elevator, et al.) and We do not pay attention to the activity of pedestrians in the static. Reference[b] proposed a threshold-based classification algorithm for carrying phone mode without experimental result and reference[c] proposed the Finite State Machine (FSM) to classify the smartphone mode with a classification accuracy of more than 89%。We all cite them in the paper, as shown at the last sentence of the second paragraph in section 1. Reference [d] uses the wearable sensor to present a comparison of the navigation accuracy between different wearable placements ( the feet and the lower back), which is different from ours.

Point 4:In Sec. 2.3. Scenario Recognition Algorithm, does the recognition results (2.3.2) of the smartphone mode affect the recognition of motion mode (2.3.3)? If so, how?

Response 4: The relationship between smartphone mode and motion mode is analyzed in Sec. 2.1. According to the analysis, we know that smartphone modes have little influence on motion mode recognition. On the contrary, motion modes have a great influence on smartphone mode recognition. And during scenario recognition, we recognize the motion mode first. And when it is determined, the smartphone mode is recognized secondly. So in Sec. 3.2 and Sec. 3.3 we will not analyze the affection between smartphone mode and motion mode.

Point 5: In Sec. 3.3, Figure 3, how do you obtain the results from other algorithms [8, 9, 30-32]? By reimplementing them? Or directly from the papers?

Response 5: In Sec. 3.3, Figure 3, we directly obtain the results from the papers[9、10、32-34]. As we can not achieve the accuracy in those papers by re-implementing them, we directly use the results from the papers.

Reviewer 2 Report

This paper presents A Novel Algorithm for Scenario Recognition Based on MEMS Sensors of smartphone. The research question is important and interesting, the method is proper and justified by some experiments. However, there are still some issues that should be addressed: 

1n and N appear many times in the article and represent different meanings, for example Line 135, Line 148, Line 156, Line 193, etc., which is confusing. 

2. Many thresholds are used for scene recognition, but no specific values are given in the experiments. 

3. From Table 1, the authors conclude that the smartphone mode is recognized firstly, and the motion mode is recognized secondly. But the reasons are not sufficient. Please explain it in detail. 

4. Are the names Number of peaks, Number of Wave and others in table 2 correct?

Author Response

Point 1: n and N appear many times in the article and represent different meanings, for example Line 135, Line 148, Line 156, Line 193, etc., which is confusing. 

 Response 1: The parameter n in Line 136(origin Line 135), Line 148 are the same parameter. N in Line 156(origin Line 157) is the the sampling length of data, and we have added “sampling” in this Line. n in Line 191(origin Line 193), Line 119, Table 2 ,etc. are different parameters and we use other parameters to replace them. For example, the parameter n in Line 120 is replaced by nk and n in Table 2 is replaced by .

Point 2: Many thresholds are used for scene recognition, but no specific values are given in the experiments. 

 Response 2: The threshold of smartphone mode and motion mode in the experimental are added in Table 8 of Sec.3.1, which is shown as follows:

Table 8. The threshold of smartphone mode and motion mode

Smartphone mode

Motion mode

 Threshold

Number

 Threshold

Number

2

1.5

1.5

0.9

10

0.9

1

0.9

Thre1

6

0.9

Thre2

6

14

Thre3

8

0.1

-0.1

Point 3: From Table 1, the authors conclude that the smartphone mode is recognized firstly, and the motion mode is recognized secondly. But the reasons are not sufficient. Please explain it in detail. 

 Response 3: In order to better explain the analysis process, we re-described it and adjusted the distribution in Table 1. The analysis is shown as follows:

To analyze the decoupling correlation different scenario categories, we set. Where i and u are the smartphone mode, j and v are the motion mode. and  are two kind of scenario. To get the analysis result we need to analyze 3 situations as follows:

The test results are shown in Table 1, which gives the correlation calculation results of a total of 9 scenarios composed of 3 motion modes and 3 smartphone modes. The raw data include GNSS sensor, accelerometer, gyroscope, magnetometer, barometer and Bluetooth. Their parameters are shown in Table 7.

Table 1. Pearson correlation coefficients of different motion modes and different smartphone modes

Wa

US

UE

Calling

Texting

Cloths pocket

Calling

Texting

Cloths pocket

Calling

Texting

Cloths pocket

Wa

Calling

0.77

-

-

-

-

-

-

-

-

Texting

0.56

0.64

-

-

-

-

-

-

-

Cloths pocket

0.55

0.53

0.77

-

-

-

-

-

-

US

Calling

0.35

0.18

0.21

0.68

-

-

-

-

-

Texting

0.27

0.34

0.25

0.55

0.80

-

-

-

-

Cloths pocket

0.27

0.24

0.33

0.51

0.52

0.67

-

-

UE

Texting

0.32

0.21

0.23

0.26

0.26

0.21

0.71

-

-

Cloths pocket

0.21

0.33

0.28

0.18

0.33

0.25

0.52

0.86

-

Texting

0.21

0.25

0.35

0.22

0.15

0.31

0.65

0.55

0.72

Wa is Walking, US is upstairs, UE is up elevator.

According to the test results in Table 1, the decoupling correlation can be summarized as follows:

From formula(7) we know that the correlation between different motion modes has a certain correlation under the same smartphone mode. And In the case of different smartphone modes, the correlation of the same motion mode is greater than 0.5. The correlation between different smartphone modes and different motion modes is very low, which is less than 0.3. So smartphone modes have little influence on motion mode recognition. On the contrary, motion modes have a great influence on smartphone mode recognition.

According to the above analysis, during scenario recognition, we can recognize the motion mode first. And when it is determined, the smartphone mode is recognized secondly.

Point 4:  Are the names Number of peaks, Number of Wave and others in table 2 correct?

Response 4: The feature names in Table 2 refer to the description of reference [24]. But description of Number of peaks and Number of Wave is not correct, and we rename them as Peaks and Wave. And the zero-crossing rate is re-defined in this paper.

Round 2

Reviewer 1 Report

All my concerns regarding this paper have been clarified in this revised paper. I think this paper is now acceptable for publication.